# Preventive Effect of Cardiotrophin-1 Administration before DSS-Induced Ulcerative Colitis in Mice

**DOI:** 10.3390/jcm8122086

**Published:** 2019-12-01

**Authors:** Ana I. Sánchez-Garrido, Vanessa Prieto-Vicente, Víctor Blanco-Gozalo, Miguel Arévalo, Yaremi Quiros, Daniel López-Montañés, Francisco J. López-Hernández, Antonio Rodríguez-Pérez, José M. López-Novoa

**Affiliations:** 1Department of Gastroenterology, University Hospital of Salamanca, 37007 Salamanca, Spain; ana_garridos@hotmail.com (A.I.S.-G.); vanessapv_78@hotmail.com (V.P.-V.); arodriguezpe@gmail.com (A.R.-P.); 2Institute for Biomedical Research of Salamanca (IBSAL), 37007 Salamanca, Spain.; marevalo@usal.es (M.A.); danielmontanes923@gmail.com (D.L.-M.); flopezher@usal.es (F.J.L.-H.); 3Bio-inRen S.L. Faculty of Medicine, Campus Miguel de Unamuno, 37007 Salamanca, Spain; vblanco@usal.es (V.B.-G.); yaremi@usal.es (Y.Q.); 4Department of Human Anatomy and Histology, University of Salamanca, 37007 Salamanca, Spain; 5Department of Physiology and Pharmacology, University of Salamanca, 37007 Salamanca, Spain

**Keywords:** apoptosis, cardiotrophin-1, colon, inflammation

## Abstract

Ulcerative colitis is a relatively frequent, chronic disease that impacts significantly the patient’s quality of life. Although many therapeutic options are available, additional approaches are needed because many patients either do not respond to current therapies or show significant side effects. Cardiotrophin-1 (CT-1) is a cytokine with potent cytoprotective, anti-inflammatory, and antiapoptotic properties. The purpose of this study was to assess if the administration of CT-1 could reduce colon damage in mice with experimental colitis was induced with 5% dextran sulfate sodium (DSS) in the drinking water. Half of the mice received an i.v. dose of CT-1 (200 µg/kg) 2 h before and 2 and 4 days after DSS administration. Animals were followed during 7 days after DSS administration. The severity of colitis was measured by standard scores. Colon damage was assessed by histology and immunohistochemistry. Inflammatory mediators were measured by Western blot and PCR. CT-1 administration to DSS-treated mice ameliorated both the clinical course (disease activity index), histological damage, inflammation (colon expression of TNF-α, IL-17, IL-10, INF IFN-γ, and iNOS), and apoptosis. Our results suggest that CT-1 administration before induction of colitis improves the clinical course, tissue damage, and inflammation in DSS-induced colitis in mice.

## 1. Introduction

Ulcerative colitis (UC) is a chronic, inflammatory bowel disease that disrupts colon structure and function. UC affects mainly young patients and is characterized by alternation of acute and remission phases. UC is associated with a higher incidence of dysplasia and colorectal cancer, and can impact significantly the patients’ quality of life and working ability [1,2]. UC is most prevalent in northern Europe, especially in the United Kingdom and Scandinavia [3,4] and North America [5], whereas Asia is also expected to reach highest incidence over the next decade [6]. UC affects the colonic mucosa and submucosa with widespread superficial ulceration, destruction of epithelial architecture and integrity, crypt loss, submucosal edema, cellular infiltration, and intense inflammation [7]. Although the precise mechanisms are not yet fully understood, several mediators, including chemotactic peptides and pro-inflammatory cytokines are known to be involved [8].

Current therapies for UC include many agents targeting different pathways involved in the induction of the disease. These agents comprise 5-aminosalicylic acid (5-ASA) precursors, such as Sulfasalazine, Mesalamine, and Budesonide, systemic (oral or intravenous) corticosteroids, thiopurines with steroid-sparing effects such as Azathioprine and 6-mercaptopurine, anti-TNF-α antibodies such as Infliximab, Adalimumab, and Golimumab, calcineurin inhibitors such as cyclosporine, anti-integrin antibodies such as Vedolizumab, Janus kinase inhibitors such as Tofacitinib and other similar molecules, and many combinations thereof. All these therapeutic approaches, their clinical use and side effects have been evaluated in several recent reviews [9,10,11], as well as briefly discussed in the Discussion section of this article. All these reviews suggest that additional treatments focused on new targets are necessary due to the serious limitations of available options.

In the present study, we assessed the utility of cardiotrophin-1 (CT-1) in an experimental model of dextran sulfate sodium (DSS)-induced colitis in mice. CT-1 is a member of the interleukin 6 (IL-6) family of cytokines, also including many other cytokines [12]. CT-1 binds to and signals through a specific membrane receptor complex containing glycoprotein 130 (gp130) and leukemia inhibitory factor receptor (LIFR) [13,14]. CT-1 is expressed in a variety of cell types and tissues and exerts potent anti-apoptotic effects on hepatocytes [15], cardiomyocytes [16] and neurons [17], and protective and anti-inflammatory effects in damaged hearts [18,19], livers [20,21], kidneys [22,23], and the nervous system [17]. Recently, we have reported that exogenous CT-1 attenuates experimental colitis in mice, when administered after the insult, suggesting a potential application in the treatment of ongoing relapses [24]. We now show that CT-1 is also able to partially prevent the development of DSS-induced experimental colitis in mice when administered before the insult, thus suggesting a potential additional prophylactic application.

## 2. Materials and Methods

Unless otherwise indicated, all reagents were purchased from Sigma-Aldrich (Madrid, Spain).

### 2.1. Animals

Twenty-five eight-week-old male mice purchased from Charles River Spain (Barcelona, Spain) were used. Only male littermates were used in order to obviate the interference of estrus. Animals were group-housed in a temperature and humidity-controlled room, with a 12-h dark/12-h light cycle (Experimental Animal Facility, University of Salamanca), and allowed free access to standard chow and tap water.

### 2.2. Ethical Considerations

All protocols were approved by the Animal Experimentation Ethics Committee of the University of Salamanca and were carried out in accordance with the Guide for the Care and Use of Laboratory Animals (NRC USA, 2011), the European Convention for the Protection of Vertebrate Animals Used for Experimental and Other Scientific Purposes, Council of Europe (ETS 123), and Spanish Government “Ley para el cuidado de los animales, en su explotación, transporte, experimentación y sacrificio” (Law 32/2007, November 7, 2017, BOE. 268, pp 45914-45920).

### 2.3. Experimental Model and Study Protocol

DSS administration is a most widely used animal model for the study of UC, for its simplicity, reproducibility, and uniformity [25]. The DSS model exhibits many symptoms similar to those seen in human UC, such as diarrhea, bloody feces, body weight loss, mucosal ulceration, and shortening of the colorectum [26].

C57BL/6J mice were randomly distributed into three groups:Colitis group (DSS; *n* = 10): Colitis was induced with 5% DSS (MW 36,000–50,000, MP Biomedical, Solon, OH, USA) in the drinking water through the experiment.Colitis + CT-1 treatment group (DSS + CT-1; *n* = 10): Mice received an i.v. dose of rat CT-1 (200 µg/kg) 2 h before and 2 and 4 days after starting DSS. Rat CT-1 was provided by DRO Biosystems (San Sebastian, Spain).Control group (Sham; *n* = 5): Mice received neither DSS nor CT-1.

The dose of CT-1 chosen is the lowest with significant effects in reducing the major symptoms in mice with DSS-induced colitis (according to our preliminary studies). Animals were followed during 7 days after starting DSS.

To assess the effects of DSS, potential behavioral alterations and changes in body weight were checked daily. Mice were euthanized when showing excessive suffering, appearing moribund, or weight loss ≥20%. At the end of the study, mice were anesthetized with 60 mg/kg pentobarbital and blood was obtained by heart puncture. Then, animals were perfused through the cardiac puncture with heparin containing (5 IU/mL) isotonic saline and the colon was dissected and cleaned. Colons were immediately trimmed out into pieces and some of the specimens were fixed in buffered 4% formaldehyde for 24 h for histological studies, whereas others samples were frozen in liquid nitrogen for biochemical measurements.

### 2.4. Colitis Severity Quantification

Colitis severity was quantified using the disease activity index (DAI) scoring body weight loss, stool consistency, and presence or absence of fecal blood as described previously [27].

### 2.5. Histological Studies

Fixed colon tissue specimens were dehydrated in ascending graded ethanol concentrations (70, 80, 90, 95, 95, 100, 100–40 min in each), followed by 25 min in xylene and 30 min in paraffin. For light microscopy, 3 µm sections were stained with hematoxylin and eosin. Additional 3 μm sections were processed for immunohistochemistry as previously reported [24]. In brief, sections were deparaffined in xylene and rehydrated in descending graded ethanol concentrations. Endogenous peroxidase was blocked with 3% hydrogen peroxide, followed by primary antibody incubation. Primary antibodies used are described in Table 1.

Then, sections were washed three times in PBS and incubated with the Novolink Polymer Detection ^®^ (RE7140-K, Novocastra, MA, USA), followed by reaction with 3,3′-diaminobenzidine as chromogen. Ten images per slide were captured with an optical microscope. Then, images were digitalized and the number of cells stained by the antibodies was quantified with the ImageJ software (Rasband, W.S., ImageJ, National Institutes of Health, Bethesda, MD, USA).

### 2.6. Western Blot (WB) Studies

Tissue extracts were obtained by homogenizing frozen colon samples with a tissue mixer (Ultra-Turrax T8, IKA^®^-Werke GmbH & Co. Staufen, Germany) at 4 °C in homogenization buffer (140 mM NaCl, 20 mM Tris-HCl Ph = 7.5, 0.5 M ethylenediaminetetraacetic acid –EDTA-, 10% glycerol, 1% Igepal CA-630, 1 µg/mL aprotinin, 1 µg/mL leupeptin, 1 µg/mL pepstatin A, and 1 mM phenylmethylsulphonyl fluoride) and used for Western blot analysis as previously described [28]. Proteins in the extracts were separated by electrophoresis in 10–15% acrylamide gels (Mini Protean II system, BioRad, Madrid, Spain), transferred to an Immobilon-P membrane (Millipore, Billerica, MA, USA), and subsequently incubated with primary and horseradish peroxidase (HRP)-conjugated secondary antibodies, described in Table 1. WB quantification was performed with the Image Quant software (GE-Healthcare, Madrid, Spain) after scanning the films with an Office jet 8500 scanner (Hewlett-Packard, Madrid, Spain).

### 2.7. Reverse Transcription-Polymerase Chain Reaction (RT-PCR) Studies

Total RNA was isolated from frozen colon samples using Nucleospin RNAII (Macherey-Nagel, Düren, Germany), according to the manufacturer’s instructions. Single-strand cDNA was generated with the M-MLV reverse transcriptase (Promega Biotech Ibérica, Alcobendas, Spain) from 2 μg of total RNA using poly-dT as a primer.

Quantitative RT-PCR was performed in triplicate. Each reaction was carried out in a volume of 20 μL containing 1 μL of cDNA, 400 nM of each primer, and 1x iQ Sybr Green Supermix (Bio-Rad, Madrid, Spain). PCR products were separated by electrophoresis in a 1% agarose gel and visualized using Invitrogen SYBR Safe DNA Gel Stain (Thermo Fisher Scientific, Carlsbad, CA, USA). Standard curves were obtained for each transcript to ensure exponential amplification and to rule out non-specific amplification. Gene expression was normalized to GAPDH expression. The reactions were studied with an iQ5 real-time PCR detection system (Bio-Rad, Madrid, Spain). The specific primers used for PCR are shown in Table 2.

### 2.8. Measurement of Cytokine Plasma Levels

Plasma levels of CT-1 (ELM-CT-1 RayBiotech, Norcross, GA, USA), INF-γ (RYD-MIF00; R&D Systems, Minneapolis, MN, USA) and TNF-α (MTA00B, R&D Systems) were measured using commercial ELISAs, according to the manufacturers’ instructions.

### 2.9. Data Statistical Analysis

The area under the curve (AUC) was calculated in each animal by adding the areas of contiguous trapezes. Statistical analysis was performed using the NCSS software. Data showing normal distribution are expressed as mean ± SEM. One-way or two-way analysis of variance was used to evaluate differences between groups, and Scheffe’s post-hoc test was performed for multiple group comparisons. Data without normal distribution are shown as medians. For group comparisons, the Kruskal–Wallis Z value test was used. P < 0.05 or Z > 1.96 were considered statistically significant.

## 3. Results

### 3.1. Evolution of Colitis

No animal died during the study. In the DSS group, there was a manifest decrease in motility and a loss of response to stimuli, compared with sham animals. In the DSS+CT-1 group, these signs were much less pronounced. No significant differences in weight were observed. The stool consistency score increased progressively in the DSS group during the observation period, and CT-1 partly prevented this increment (Figure 1A). The area under the curve (AUC) for stool consistency was 0 for sham animals, 4.51 ± 0.60 arbitrary units (AU) in DSS-treated animals and 2.10 ± 0.33 AU in DSS+ CT-1 mice (Z > 1.96). The fecal blood score also increased progressively in the DSS group; whereas in the animals that received DSS + CT-1, it was lower at all time points. The disease activity index (DAI) was always lower in CT-1-treated than in untreated mice (Figure 1B). The area under the curve (AUC) for DAI was 1.51 ± 0.3 for sham animals, 5.61 ± 0.61 arbitrary units (AU) in DSS-treated animals, and 3.94 ± 0.33 AU in DSS+ CT-1 mice (Z > 1.96).

### 3.2. Histological Characterization

Hematoxylin-eosin staining of colon specimens revealed that mice treated with DSS showed typical alterations of ulcerative colitis. Ulcers were focally distributed through the inner surface of the colon and invaded the entire thickness of the colon wall, even penetrating the lamina propria. In ulcerated areas, total crypt destruction and epithelial loss were seen, including goblet cells. Granulation tissue and a large transmural inflammatory infiltrate occupied the ulcerated zones (Figure 1C). Peripheral tissue surrounding colonic ulcers showed normal structure, without alterations. In mice treated with DSS plus CT-1, even though ulcers were observed, mainly in the distal colon, these were fewer than those found in the control group and their size was also smaller. In many areas, although an infiltration of inflammatory cells was occasionally detected, the epithelium of the colon mucosa appeared fairly well preserved, including goblet cells, and most of the colon had no structural alterations (Figure 1D). Hematoxylin-eosin-stained, proximal colon sections from the sham group showed intact epithelium, well defined crypt length, absence of edema or infiltrating cells in the mucosa and submucosa, and no ulcers or erosions (Figure 1E).

### 3.3. Evaluation of Colonic Inflammation

CD-68 immunohistochemistry was performed to assess mucosal monocyte/macrophage infiltration. Sham animals showed a very low number of CD-68 positive cells. A high number of CD-68 positive cells was observed in DSS-treated mice, whereas this number was significantly lower in the animals treated also with CT-1 than in those that received only DSS (Figure 2A,B).

Immunohistochemistry for inducible nitric oxide synthase (iNOS), an enzyme associated to inflammation, revealed that cells expressing iNOS were very scarce in the colonic tissue of animals from the sham group, but very abundant in the animals receiving only DSS (Figure 2C). In the animals treated also with CT-1, the number of cells expressing iNOS was significantly lower than in animals receiving DSS alone (Figure 2D). Western blot studies also demonstrated that iNOS levels in animals that received DSS were more than 5 times higher than those of sham animals and that co-administration of CT-1 significantly reduced iNOS to levels similar to those in sham animals (Figure 2E).

Inflammation was also assessed by plasma levels and by colonic gene expression of several cytokines. Mice with UC showed higher plasma TNF-α concentrations than mice in the sham group. Mice treated with DSS that also received CT-1 showed significantly lower plasma levels of TNF-α than those that only received DSS, and were similar to those in the sham group (Figure 3A). Colonic TNF-α gene expression (assessed by PCR) was significantly higher in the DSS than in the sham group, but without statistically significant differences with respect to animals that also received CT-1 (Figure 3B,C). Gene expression of IL-17 in colonic tissue was significantly higher in mice treated with DSS than in sham animals. In mice pre-treated with CT-1, colon IL-17 gene expression was significantly lower than in animals treated with DSS alone, and not significantly different from those in sham animals (Figure 3B–D). Gene expression of IFN-γ in colonic tissue was significantly higher in mice treated with DSS than in sham animals. In mice that also received CT-1 in addition to DSS, gene expression of INF-γ in colon was not significantly different from that in sham animals (Figure 3B,E). Similar results were obtained for IL-10 expression (Figure 3B–F).

NF-κB activation was assessed by quantifying the levels of phospho-p65 (pp65) as a fraction of total p65 (RelA) in colonic tissue. Animals treated with DSS had a higher pp65/p65 ratio than controls, and co-treatment with CT-1 significantly reduced the ratio to levels in controls (Figure 4A). Animals receiving DSS showed a higher pStat-3/Stat-3 ratio than sham animals. Co-treatment with CT-1 reduced the ratio pStat-3/Stat-3, but this reduction was not statistically significant when compared with the mice that received only DSS (Figure 4B).

### 3.4. Apoptosis Evaluation

Apoptosis was assessed by immunostaining through the number of cleaved caspase 3 positive cells. Representative images are shown in Figure 4C. The number of cleaved caspase 3-positive cells in the colon was very low in control animals, but significantly higher in animals receiving DSS. Animals treated with CT-1 had fewer cleaved caspase 3-positive cells than untreated animals (Figure 4D).

### 3.5. CT-1 Expression

Plasma levels of CT-1 were higher in animals receiving DSS than in control animals, but the difference was not statistically significant. Animals receiving CT-1 + DSS showed higher plasma levels of CT-1 than mice receiving DSS alone, although the difference did not reach statistical significance either (Figure 5A). CT-1 content in colonic tissue was significantly higher in DSS-treated animals than in control mice. Mice receiving CT-1 + DSS showed significantly higher level of CT-1 in the colon than animals receiving DSS alone (Figure 5B). CT-1 gene expression in colon was higher in DSS than in the sham group, without statistically significant differences. In animals that also received CT-1, CT-1 gene expression in colon was significantly higher than in the DSS group (Figure 5C). 

## 4. Discussion

UC is a chronic disease affecting the large intestine, with an increasing incidence worldwide. UC is an immune-mediated condition frequently associated with inflammation of the rectum, but often extends proximally to involve additional areas of the colon. Management of UC must include a prompt and accurate diagnosis, assessment of the patient’s risk of poor outcomes, and initiation of effective, safe, and tolerable medical therapy to induce and maintain remission [9]. The usual sequence of treatment choice starts with 5-aminosalicylates, then steroids, thiopurines, anti–tumor necrosis factor-α antibodies, Vedolizumab and, finally, surgery.

For patients with mild-to-moderate disease, 5-aminosalicylic acid (5-ASA) precursors, such as Sulfasalazine, Mesalamine, and Budesonide, have been used either by oral or rectal administration, or a combination of both. Some of these products have notable side effects such as anemia, abnormal liver tests, nausea, headaches, fever and rash, which are more frequent after treatment with Sulfasalazine than with Mesalamine and Budesonide [10,11]. In addition, in more severe cases, 5-ASA precursors show only a limited efficacy.

Corticosteroids are frequently used for the induction of remission in patients with moderate-to-severe disease. Oral steroids are used in most cases, but intravenous administration should be performed in patients with acute severe ulcerative colitis. Steroids are very effective in inducing remission but they are ineffective in maintaining remission, and steroid treatment needs to be withdrawn after long term therapy due to toxic effects. This toxicity may involve all organs and damage is frequently irreversible [29]. Furthermore, the risk of severe side effects increases when steroids are used in combination with other immunosuppressive agents.

After 5-ASA, thiopurines [i.e. the precursors azathioprine (AZA) and 6-mercaptopurine (6-MP)] are the most frequently used drugs worldwide for UC chronic treatment. Several meta-analyses suggest that AZA and 6-MP are useful to maintain corticosteroid-free clinical remission, but they are not effective in inducing relapse remission. AZA and 6-MP are slow-acting drugs, so longer treatment with steroids is necessary before the inception of their therapeutic effect ensues, thus increasing the risk of steroid toxicity. In addition, AZA and 6-MP have frequent side effects, such as elevation of plasma transaminases and leukopenia, as well as increased risk of lymphoma and non-melanoma skin malignancy [30]. Consequently, skepticism about their use in UC has emerged.

TNF-α plays a major role in gut inflammation seen in UC. Anti-TNF-α antibodies bind to TNF-α and counteract its biological effects. The most used anti-TNF-α antibodies are Infliximab, Adalimumab, and Golimumab. Treatment with these drugs, alone or in combination with steroids, induces and maintains the remission of UC [31]. However, anti-TNF-α antibodies are not useful or effective in one third of patients (i.e., “primary failures”), and in another third, they lose effect over time (“secondary failures”) [32]. Moreover, these treatments are often complicated by multiple side effects [33] and high price, leading to differential access among European countries [34].

Cyclosporine, a calcineurin inhibitor frequently used as immunosuppressant in transplantation, has also been used for the treatment of patients with steroid-refractory, severe UC with positive results, and induces remission in many patients [35]. Studies on another calcineurin inhibitor, tacrolimus, in UC are very scarce and most of them do not recommending its use. Cyclosporine is not recommended for chronic treatment because, despite over 60% of patients showing a clear improvement, most of them have to be subject to colectomy after a prolonged treatment (5–7 years) [35].

Anti-integrin antibodies are also employed for UC treatment. Integrins regulate the process of leucocyte migration from the blood to the gut. Vedolizumab is a fully humanized recombinant monoclonal, anti-integrin antibody that decreases leucocyte migration to the intestine. Vedolizumab was the first anti-integrin antibody approved for the treatment of UC, and it has been reported to be effective based on its successful ability to induce and maintain relapse remission in patients with moderate-to-severe UC. The most frequent side effects of Vedolizumab include common cold symptoms, headache, joint pain, nausea, fever, upper respiratory tract infection, fatigue, rash, and limbs pain [36].

Tofacitinib is a Janus kinase (JAK) inhibitor useful to treat adults with moderate-to-severe UC. JAK activation initiates a signaling cascade involved in many biological effects, including inflammation. JAK is a tyrosine kinase activated by many factors, including interleukine-6 (IL-6), that plays a major role in UC-associated inflammation. IL-6 binds the IL-6r-Gp130 receptor complex and activates the JAK. Once activated, JAK proteins change their conformation, dimerize, phosphorylate, and activate their primary substrates, the signal transducer and activator of transcription (STAT) proteins. Tyrosine-phosphorylated STAT proteins homo- or hetero-dimerize and translocate to the nucleus, where they interact with coactivators and bind to specific regulatory elements in the promoter regions of thousands of different genes, including those codifying proteins involved in the inflammatory process. [37]. Other JAK inhibitors such as filgotinib, upadacitinib, TD-1473, peficitinib, and Pf-06651600/Pf-06700841 have been recently developed, but only data from phase I and II clinical trials are available for most of them [38]. Although Tofacitinib is a useful addition to the arsenal of drugs available for UC treatment, many adverse effects have been reported including severe bacterial, fungal, and viral infections but also by opportunistic pathogens such as herpes zoster [39]. In addition, some studies suggest that Tofacitinib should not be used in combination with immunosuppressants such as a thiopurines or calcineurin inhibitors [39].

Another therapeutic approach for UC treatment is the use of mesenchymal stem cells, although the development of this treatment is in a very preliminary phase [40].

Because, as described above, most drugs used for the treatment of UC become progressively ineffective over time and/or present or show severe side effects, new therapeutic strategies are necessary to target the multiple mechanisms responsible for colon damage. In a previous study [24], we demonstrated that CT-1 administration on established experimental colitis reduces symptom severity, colon inflammation intensity and histological damage. In this article, we evaluated if CT-1 was also effective in decreasing the severity of colitis when the symptoms were not yet clearly noticeable. In the clinical practice, this would allow CT-1 use when relapses are foreseeable on the basis of analytical data.

In the whole, this study reveals that CT-1 administration before the induction of experimental UC with DSS ameliorates both the clinical course of the disease, the severity of colon inflammation, and the histological damage of the colon. Protection appears to be provided by several mechanisms. Similarly, as in other organs including the lungs [41], heart [42], liver [43], and kidneys [23], CT-1 has a potent anti-inflammatory effect in this model, as revealed by lower plasma level of TNF-α, lower expression of IL-1β, IL-17 in the colon, less pronounced increased expression of iNOS, and lower colon infiltration of monocytes/macrophages. CT-1 also ameliorates colonic apoptosis in this model, as in other organs and cells such as isolated myocardiocytes [44,45], liver [43,46,47], pancreatic beta cells [48], neurons [49], and kidney epithelial cells [23]. When comparing the results of this prophylactic study with those from the curative strategy previously published [24], both approaches ameliorate the course of disease and preserve colon tissue, although colon apoptosis and most indicators of inflammation are further minimized by the preventive strategy with CT-1. With a therapeutic aim, both studies provide a potential tool for the management of ongoing UC as a complement to prophylaxis. Because prevention of relapses poses a more complicated intervention involving the monitoring of early signs and symptoms and early biomarkers such as fecal calprotectin [50,51], patient handling and outcome might benefit from the combined handling incorporating the preventive and curative approaches.

Despite a similar outcome, the preventive and curative mechanisms behind CT-1 action might be, at least partly, different. In the case of the preventive effect, CT-1 might precondition the colon to oppose damage; whereas in the curative action, CT-1 might reverse damage mechanisms. This is hypothetically inferred from the different behavior of specific mediators. In the preventive model, CT-1 significantly moderates the increase in colonic activation of STAT-3 and NF-κB by DSS, whereas in the curative model, CT-1 has no effect on or, if anything, increases colonic STAT-3 and NF-κB activation [24]. This is consistent with the dual role of NF-κB in inflammation [52,53]. During the initiation phase, NF-κB contributes to the inflammatory process by increasing the expression of proinflammatory chemokines and adhesion molecules, and promotes epithelial cell proliferation and survival, whereas during the resolution phase, NF-κB favors leukocyte apoptosis [52,53]. This suggests that CT-1 may also act upstream of inflammation, likely by preventing tissue damage and cell death. Preventive and curative data on the role of NF-κB would be in agreement with inflammation being the consequence of colon injury, as previously hypothesized [52]. In the preventive scenario, CT-1 would prevent tissue damage and, consequently, the activation of NF-κB and inflammation. In the curative scenario, some degree of damage would occur before CT-1 administration, which would give rise to an inflammatory response, and some colon apoptosis. Apoptosis progression would be prevented by CT-1 from the moment of treatment inception, in the presence of an established inflammatory response, characterized by activation of NF-κB for the resolution phase. Similarly, a reference indicator of inflammation, namely the plasma level of TNF-α, is also congruent with this hypothesis. In fact, plasma TNF-α in CT-1-treated mice with ulcerative colitis is almost normal (i.e., undistinguishable from the controls), whereas it was markedly lower to that found in untreated mice with colitis.

Clinical application of CT-1 on UC is expected to be reasonably feasible because CT-1 has been already authorized for use in humans by several national and international agencies. The US Food and Drug Administration (FDA) has granted CT-1 the status of “orphan drug” for acute liver failure management (designation request 11-3507) and for the prevention of liver damage prevention during liver transplantation (designation request 07-2449). Orphan drug status has also been granted to CT-1 by the European Medicines Agency (EMA) for the prophylaxis of ischemia/reperfusion damage during transplantation of solid organs (EU/3/06/396). Furthermore, a trial to evaluate CT-1 safety, tolerability, and pharmacokinetics in healthy volunteers has been registered at ClinicalTrials.gov (https://clinicaltrials.gov/; identifier NCT01334697, accessed on October 24, 2019). It has been reported that long term administration or high dosage of CT-1 can induce structural (fibrosis) and functional injury in the kidneys and heart [54]. However, when CT-1 is administered in low doses or for short periods of time, these undesirable effects do not occur [55]. For these reasons, we suggest that CT-1 therapy might be potentially used as an acute treatment (4–6 days) for both reducing the severity of UC symptoms and to prevent clinically predicted relapses.

## 5. Conclusions

Our data demonstrate that CT-1 administration before the induction of UC with DSS, ameliorates both the clinical course and the histological damage in the colon. This effect seems to be mediated by direct attenuation of inflammation and apoptosis, and activation of the Stat-3 and NF-κB pathways. Thus, acute therapy with CT-1 could also be envisaged as a potential, new therapeutic tool against UC relapse.

## Figures and Tables

**Figure 1 jcm-08-02086-f001:**
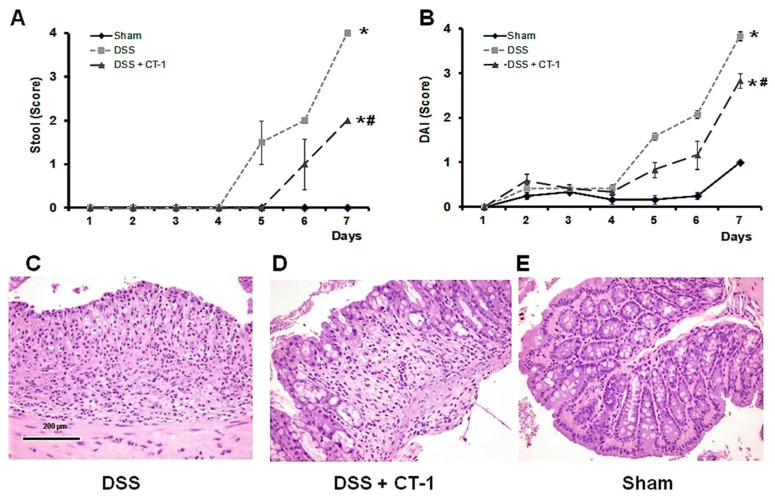
Effect of CT-1 on clinical and histological damage in dextran sulfate sodium (DSS)-induced colitis in mice. (**A**) Stool consistency score. (0: normal feces; 4: massive hemorrhage). (**B**) Disease activity index (DAI). (**C**–**E**) Representative histology images of colonic damage. Sections (4 µm) were stained with hematoxylin-eosin. Bar: 200 µm. Values are expressed as mean ± SEM (panels A and B: Sham, *n* = 4; DSS, *n* = 8; DSS + CT-1, *n* = 8.). *: *p* < 0.05 vs. Sham group; #: *p* < 0.05 vs. DSS group.

**Figure 2 jcm-08-02086-f002:**
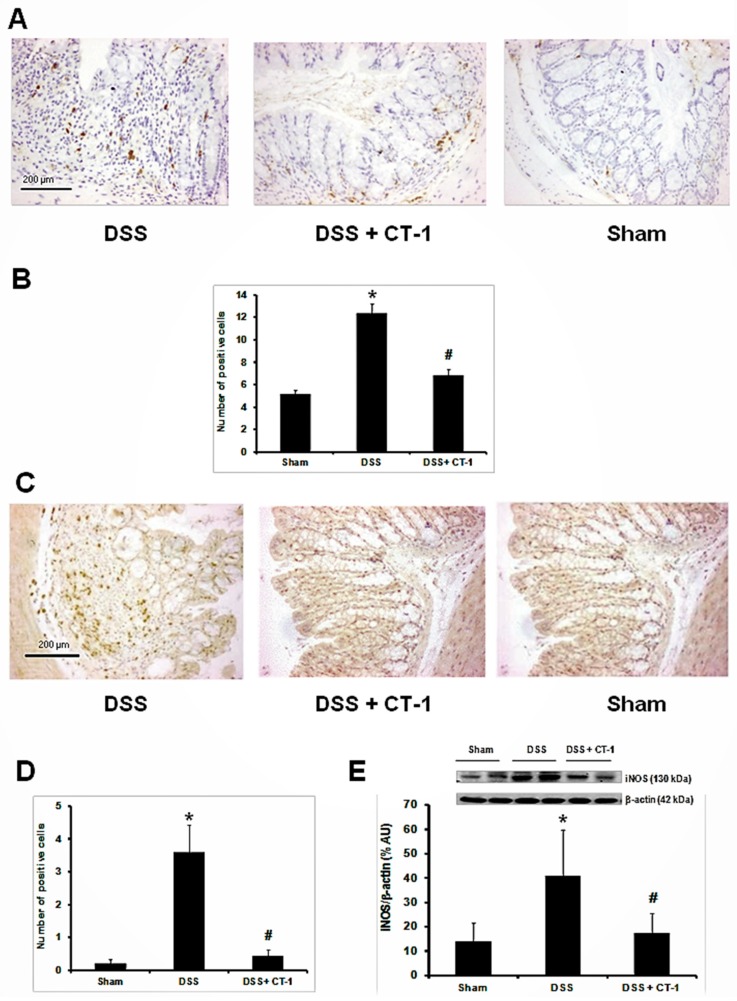
Effect of CT-1 on colonic inflammation in dextran sulfate sodium (DSS)-induced colitis in mice. (**A**) Representative images of CD-68 staining in colon. Bar: 200 µm. (**B**) Number of CD-68 positive cells. (**C**) Representative images of iNOS staining in colon. Bar: 200 µm. (**D**) Number of cells positive for iNOS staining. (**E**) Western blot analysis of iNOS and β-actin levels in colon tissue homogenates, expressed as % arbitrary units (% AU). Values are expressed as mean ± SEM. The number of slides quantified per group in panels A to D was as follows: Sham, *n* = 40; DSS, *n* = 60; DSS + CT-1, *n* = 60. In panel E: Sham, *n* = 4; DSS, *n* = 5; DSS + CT-1, *n* = 5. *: Z > 1.96 vs. Sham group; #: Z > 1.96 vs. DSS group.

**Figure 3 jcm-08-02086-f003:**
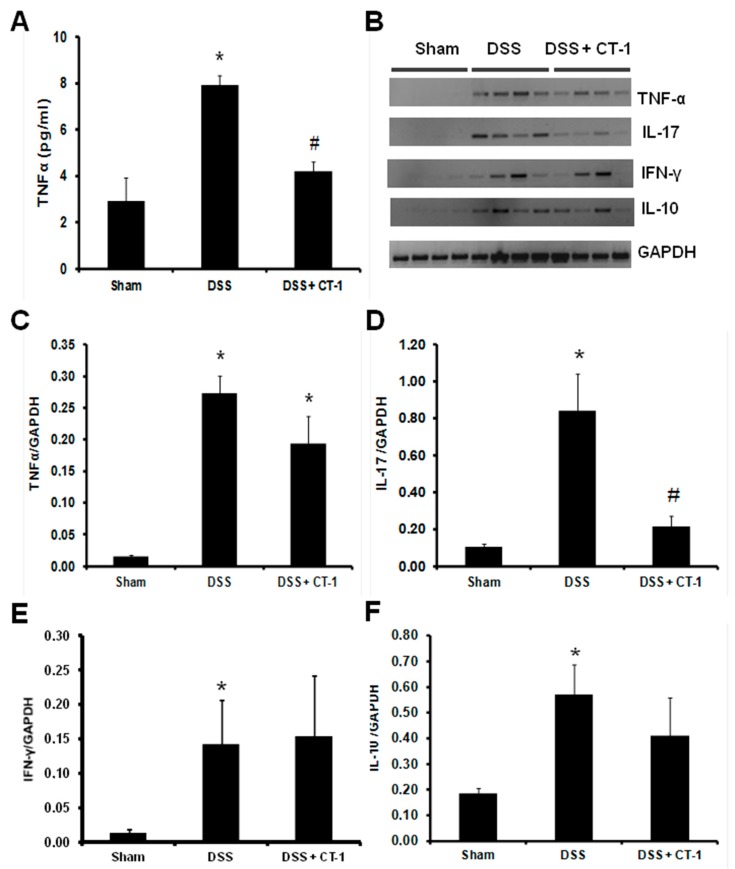
Effect of CT-1 on colon cytokine production in dextran sulfate sodium (DSS)-induced colitis in mice. (**A**) Plasma TNF-α levels expressed as pg/mL. (**B**) PCR analysis of TNF-α, IL-17, IFN-γ, IL-10, and GAPDH levels in colon tissue homogenates. (**C**) TNF-α/GAPDH quantification. (**D**) IL-17/GAPDH quantification. (**E**) IFN-γ/GAPDH quantification. (**F**) IL-10/GAPDH quantification. Values are expressed as mean ± SEM (Sham, *n* = 4; DSS, *n* = 8; DSS + CT-1, *n* = 8). *: *p* < 0.05 vs. Sham group; #: *p* < 0.05 vs. DSS group.

**Figure 4 jcm-08-02086-f004:**
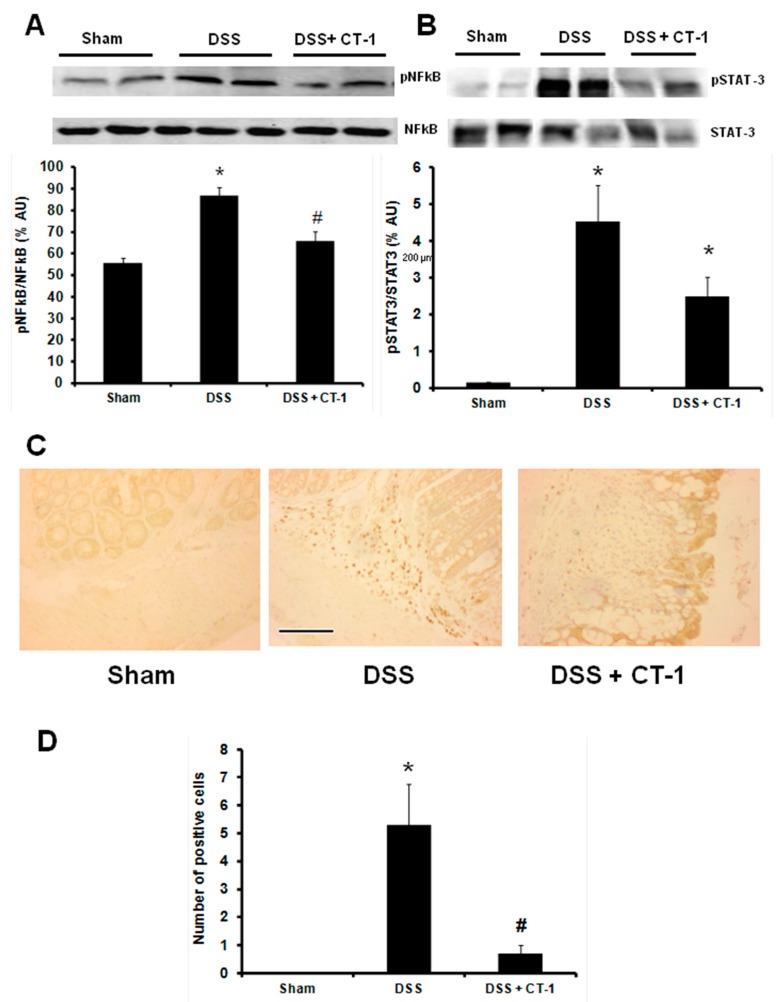
Effect of CT-1 on NF-κB activation, STAT-3 activation and apoptosis in dextran sulfate sodium (DSS)-induced colitis in mice. (**A**) Western blot analysis of N-FκB and NF-κB activation (pNF-κB) levels in colon tissue homogenates and expressed as % arbitrary units (% AU). (**B**) Western blot analysis of STAT-1 and STAT-3 activation (pSTAT-3) levels in colon tissue homogenates and expressed as % arbitrary units (% AU). (**C**) Representative images of cleaved caspase 3 staining in colon; Bar: 200 µm. (**D**) Number of cleaved caspase 3 positive cells. Values are expressed as mean ± SEM (panels A and B: Sham, *n* = 8; DSS, *n* = 10; DSS + CT-1, *n* = 10). Slides quantified per group in panels C and D: Sham, *n* = 40; DSS, *n* = 60; DSS + CT-1, *n* = 60. *: *p* < 0.05 vs. Sham group; #: *p* < 0.05 vs. DSS group.

**Figure 5 jcm-08-02086-f005:**
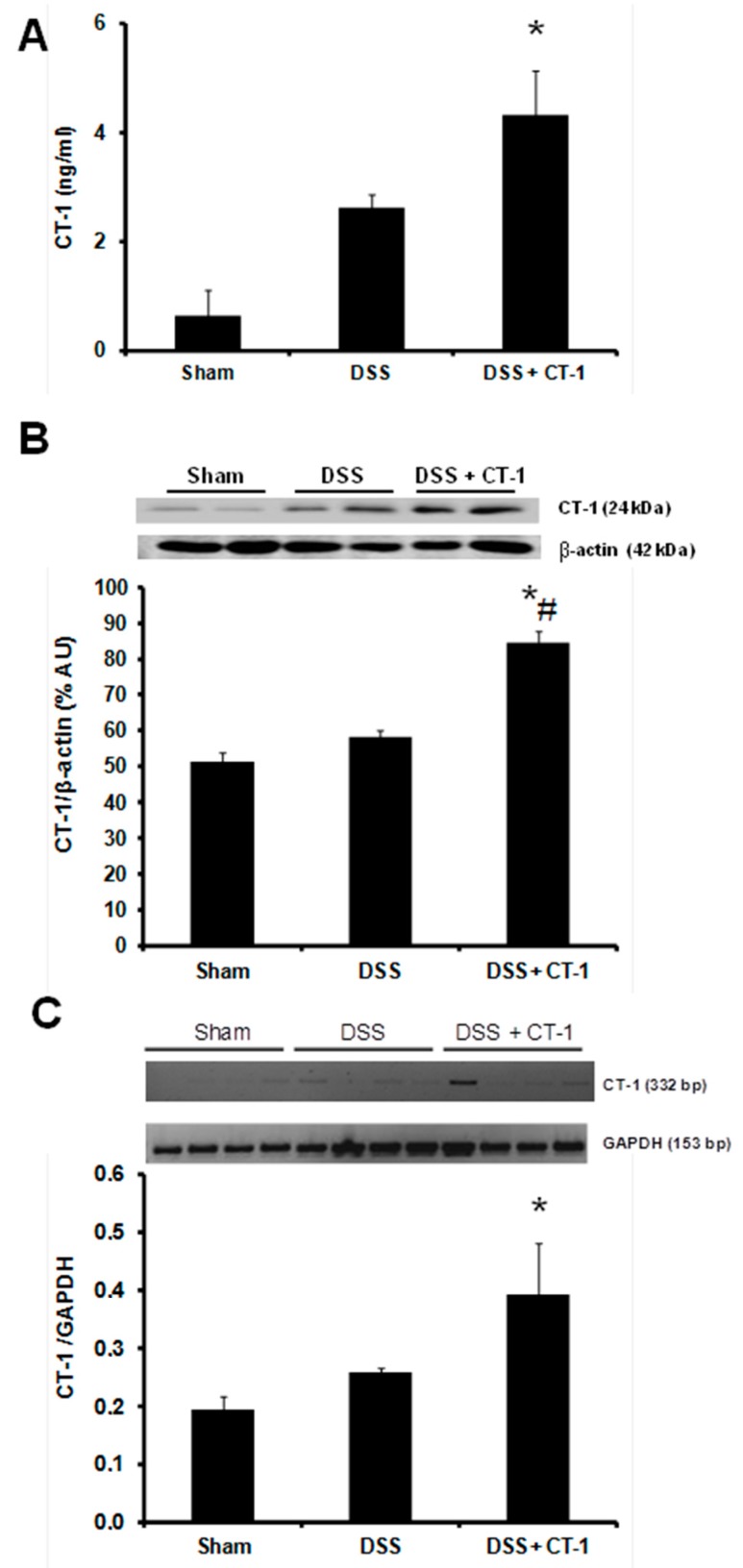
Effect of CT-1 on plasma levels and colonic production of CT-1 in dextran sulfate sodium (DSS)-induced colitis in mice. (**A**) Plasma CT-1 levels. Values are expressed as mean ± SEM (Sham, *n* = 3; DSS, *n* = 4; DSS + CT-1, *n* = 4). (**B**) Western blot analysis of CT-1 and β-actin levels in colon tissue homogenates, expressed as % arbitrary units (% AU). Values are expressed as mean ± SEM (Sham, *n* = 8; DSS, *n* = 10; DSS + CT-1, *n* = 10). (**C**) PCR analysis of CT-1 and GAPDH levels in colon tissue homogenates and CT-1/GAPDH quantification. Values are expressed as mean ± SEM (Sham, *n* = 4; DSS, *n* = 4; DSS + CT-1, *n* = 4). *: *p* < 0.05 vs. Sham group; #: *p* < 0.05 vs. DSS group.

**Table 1 jcm-08-02086-t001:** Antibodies used for Western blot and immunohistochemistry studies.

*Western Blot*		
**Primary Antibody**	**Catalog Number**	**Supplier**
iNOS	#2977	Cell Signaling Technology, Inc.
CT-1	MAB438	R&D Systems
NF-kB	#3034	Cell Signaling Technology, Inc.
pNF-kB	#3031	Cell Signaling Technology, Inc.
STAT3	#9132	Cell Signaling Technology, Inc.
pSTAT3	#9145	Cell Signaling Technology
Cleaved caspase 3	#9661	Cell Signaling Technology, Inc.
**Secondary Antibody**	**Catalog Number**	**Supplier**
Anti-rabbit IgG-HRP	4052-05	Southern Biotech.
Anti-IgG rat IgG-HRP	sc-2006	Santa Cruz Biotechnology
Anti-mouse IgG-HRP	1034-05	Southern Biotech.
**Immunohistochemistry**	**Catalog Number**	**Supplier**
CD68	M0814	Dako Diagnósticos, Spain
iNOS	sc-651	Santa Cruz Biotechnology
Cleaved Caspase 3	#9661	Cell Signaling Technology
**Suppliers**		
Cell Signaling Technology, Inc., Danvers, Massachusetts, USA		
R&D Systems, Minneapolis, Minnesota, USA		
Santa Cruz Biotechnology, CA, USA		
Southern Biotech., Birmingham, USA		
Dako Diagnósticos, Barcelona, Spain		

**Table 2 jcm-08-02086-t002:** Sequences of the primers used and conditions of PCR studies.

Gen	*Primer*	Sequence (5′-3′)	Amplicon (pb)	Tm (°C)
TNF-α	*Fw*	AGCACAGAAAGCATGATCCG	212	60
	*Rv*	CTGATGAGAGGGAGGCCATT		
IFN-γ	*Fw*	TACACACTGCATCTTGGCTTTG	128	57.5
	*Rv*	CTTCCACATCTATGCCACTTGAG		
IL-10	*Fw*	ATGCTGCCTGCTCTTACTGACTG	216	58.8
	*Rv*	CCCAAGTAACCCTTAAAGTCCTGC		
IL-17	*Fw*	GCTCCAGAAGGCCCTCAGA	142	58.8
	*Rv*	AGCTTTCCCTCCGCATTGA		
GAPDH	*Fw*	GTCGGTGTGAACGGATTTG	153	55.9
	*Rv*	GAATTTGCCGTGAGTGGAGT		

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
