# Peer review of "Preventive Effect of Cardiotrophin-1 Administration before DSS-Induced Ulcerative Colitis in Mice"

_jcm, 2019, doi:10.3390/jcm8122086_

Round 1

Reviewer 1 Report

In the manuscript “Preventive effect of Cardiotrophin-1 administration before DSS-induced ulcerative colitis in mice” by Sanchez-Garrido et al., the authors show that intravenous injection of cardiotrophin-1 in mice 2 hours prior to and after DSS treated ameliorated colitis. Clinical disease index, Western blot analysis and PCR analysis all showed marked decrease in colitis upon cardiotrophin-1 (CT-1) treatment. The current study is a follow up of a prior study by the same group (reference 24) showing that CT-1 can be used to decrease colitis in DSS treated mice. The data is straightforward and the results are clear. However, the only main question that I have is the significance of this study compared to the study in REF 24. In the prior study (REF 24), the authors conducted a more thorough analysis of the therapeutic effect of CT-1 in DSS mice – using CT-1 knockout mice. In addition, the authors also administered exogenous CT-1 into DSS-treated wild-type mice. Please state in the introduction/discussion, the difference in significance between the current study and the results from REF 24. For example, "cardiotrophin was administered prior to manifestation of colitis". Other minor comments are listed below.

In the title, suggest that “Cardiotrophin-1” be changed to “cardiotrophin-1” The title implies that cardiotrophin-1 pre-administration in DSS colitis prevented colitis. I fully agree that cardiotrophin-1 treatment diseased colitis in the DSS model. However, the cardiotrophin-1 was given 2 hours prior to DSS treatment and 2 and 4 days after DSS treatment. Please state the significance of this manuscript more clearly (see my opening comments in the evaluation above) Line 89. Please remove “41” at the end of the sentence. Line 113. Table. The catalog number and supplier column is haphazard. Please align the items. Line 25 and Line 145 (Table). IFN-g and INF-g should be written in a consistent manner. Figure 2A, 2C. In both figure panels, the resolution and color of the IHC images are difficult to interpret. Please provide a high resolution image and a clearer image. Also, Figure 4C. Line 210. “figure 2E” should be changed to “Figure 2E) Line 237. “NF B activation” should be changed to “NF-kB activation”. Line 258. Figure 5B. In the WB image, “beta-actina” should be changed to “beta-actin” Please state the source of the cardiotrophin-1 used in this study. Was it from Sigma?

Reviewer 2 Report

This study is about the preventive effect of cardiotrophin-1(CT-1) on DSS-induced UC in mice. The manuscript is well-written, but there are several issues which are required to be addressed.

Comments:

In the animal study, there are one concentration (200 ug/kg) was used. There is no explanation why this concentration was chosen, and multiple concentrations has to be used to see the dose-dependency. The data is not enough to elucidate the effect of CT-1 on UC. There are so many studies to look at the effect of compounds on DSS-induced UC and tested the regulation of NF-kB, STAT3, and cytokines. To elucidate the significant effect of CT-1, the additional experiments for evaluating the mechanism of action has to be performed. In the discussion, lots of part is about reviewing the current drugs on UC. Rather than reviewing them, discussion about the significance of CD-1's effect, possible mechanisms, advantage of CT-1 compared to others etc has to be discussed. Quantification graph of blot does not need to separated. For example, Fig. 3C~3F is quantification data of Fig. 3B.  Fig. 3C~3F has to be combined into Fig. 3B.

Round 2

Reviewer 1 Report

This reviewer is satisfied with the changes made by the authors.

Author Response

All the errors of grammar, word choice or sentence construction along the full manuscript have been reviewed by an English-speaking colleague. In the Abstract, the sentence "some mice..." has been changed to "A half of mice" In the abstract, "experimental UC" has been replaced by "experimental colitis" NF-kB has been always written with the hyphen between F and k 
